# Prevalence and molecular characterization of *Entamoeba moshkovskii* in diarrheal patients from Eastern India

**Sanjib K. Sardar**[1]**, Ajanta Ghosal**[1]**, Tapas Haldar**[1]**, Maimoon Maruf**[1]**, Koushik Das**[1,2]**, Yumiko Saito-Nakano**[3]**, Seiki Kobayashi**[4]**, Shanta Dutta**[5]**, Tomoyoshi Nozaki**[6]**, Sandipan Ganguly**[1] *

**1** Division of Parasitology, ICMR-National Institute of Cholera and Enteric Diseases (ICMR-NICED), Kolkata, India, **2** Department of Allied Health Sciences, School of Health Sciences, University of Petroleum and Energy Studies, Dehradun, India, **3** Department of Parasitology, National Institute of Infectious Diseases (NIID), Tokyo, Japan, **4** Department of Infectious Diseases, Keio University School of Medicine, Tokyo, Japan, **5** Division of Bacteriology, ICMR-National Institute of Cholera and Enteric Diseases (ICMR-NICED), Kolkata, India, **6** Department of Biomedical Chemistry, School of International Health, Graduate School of Medicine, The University of Tokyo, Tokyo, Japan

* sandipanganguly@hotmail.com

**Data Availability Statement:** Representative sequences obtained in this study were deposited in GenBank under the accession numbers ON965383 - ON965450.

## Abstract

### Background

Importance of the amphizoic amoeba *Entamoeba moshkovskii* is increasing in the study of amoebiasis as a common human pathogen in some settings. Limited studies are found on the genetic and phylogenetic characterization of *E. moshkovskii* from India; hence remain largely unknown. In this study, we determined the prevalence and characterized the *E. moshkovskii* isolates in eastern India.

### Methods

A three-year systemic surveillance study among a total of 6051 diarrhoeal patients from ID Hospital and BC Roy Hospital, Kolkata was conducted for *E. moshkovskii* detection via a nested PCR system targeting 18S rRNA locus. The outer primer set detected the genus *Entamoeba* and the inner primer pair identified the *E. moshkovskii* species. The 18S rRNA locus of the positive samples was sequenced. Genetic and phylogenetic structures were determined using DnaSP.v5 and MEGA-X. GraphPad Prism (v.8.4.2), CA, USA was used to analyze the statistical data.

### Result

4.84% (95%CI = 0.0433–0.0541) samples were positive for *Entamoeba* spp and 3.12% (95%CI = 0.027–0.036) were infected with *E. moshkovskii*. *E. moshkovskii* infection was significantly associated with age groups ($X^2 = 26.01$, P<0.0001) but not with gender (Fisher's exact test = 0.2548, P<0.05). A unique seasonal pattern was found for *E. moshkovskii* infection. Additionally, 46.56% (95%CI = 0.396–0.537) were sole *E. moshkovskii* infections and significantly associated with diarrheal incidence ($X^2 = 335.5$, df = 9; P<0.0001). Sequencing

**Funding:** The National Institute of Infectious Diseases (NIID), Tokyo, Japan, and the Indian Council of Medical Research (ICMR), Government of India, provided grant funding to Dr. Sandipan Ganguly. Sanjib K. Sardar received financial support through the ICMR Fellowship. The funders had no role in study design, data collection and analysis, decision to publish, or preparation of the manuscript.

**Competing interests:** The authors have declared that no competing interests exist.

revealed that the local *E. moshkovskii* strains were 99.59%-100% identical to the prototype (GenBank: KP722605.1). The study found certain SNPs that showed a correlation with clinical features, but it is not necessarily indicative of direct control over pathogenicity. However, SNPs in the 18S rRNA gene could impact the biology of the amoeba and serve as a useful phylogenetic marker for identifying pathogenic *E. moshkovskii* isolates. Neutrality tests of different coinfected subgroups indicated deviations from neutrality and implied population expansion after a bottleneck event or a selective sweep and/or purifying selection in co-infected subgroups. The majority of $F_{ST}$ values of different coinfected subgroups were <0.25, indicating low to moderate genetic differentiation within the subgroups of this geographical area.

## Conclusion

The findings reveal the epidemiological significance of *E. moshkovskii* infection in Eastern India as the first report in this geographical area and expose this species as a possible emerging enteric pathogen in India. Our findings provide useful knowledge for further research and the development of future control strategies against *E. moshkovskii*.

### Author summary

Although *Entamoeba* genus consists of many different species, not all are pathogenic. *Entamoeba histolytica* is the only species recognized as a definite pathogen associated with intestinal and extraintestinal infections. Nowadays, the importance of other morphologically identical *Entamoeba* species, like amphizoic *E. moshkovskii* is increasing, as it has been reported in human patients over the years from different countries. Investigations into the pathogenic potential of *E. moshkovskii* are ongoing, and only limited studies have been conducted on genetic characterization from India. This study reveals the epidemiology and population structure of *E. moshkovskii* in diarrhoeal patients from eastern India. Our study showed that the prevalence of *E. moshkovskii* in diarrhoeal patients is higher than *E. histolytica* in the studied region. In addition, there was a unique seasonal pattern, found for the infection. The high prevalence rate of sole infection in patients suggests that it is one of the definite etiological agents for diarrhoeal disease in eastern India. The further study identified two SNPs in the 18S rRNA locus, significantly associated with the sole infection cases and hence they can be regarded as a marker for identifying pathogenic isolates of *E. moshkovskii*. Many novel genotypes were obtained in the study indicating high genetic diversity in *E. moshkovskii* population conferring adaptability in a changing environment. Further research is needed to ensure proper control measures for this infection for better health care.

## Introduction

Amoebiasis, an infection by enteric protozoa, most commonly *Entamoeba histolytica* is globally considered a potentially severe and life-threatening infection [1]. The motile form of the parasite, the trophozoite inhabits the lumen of the large intestine, where it multiplies and differentiates into a resistant form. These cysts are released into the environment [1,2]. Consecutively, this resistant form is responsible for the transmission of the infection to another host via the faecal-oral route. Together with the pathogenic species *E. histolytica*, the genus *Entamoeba*

includes many other species that reside in the human intestinal lumen, namely *E. dispar*, *E. moshkovskii*, *E. bangladeshi*, *E. polecki*, *E. coli*, and *E. hartmanni* [3]. In many cases, infections with the trophozoites of *Entamoeba* spp. can result in harmless colonization in the intestinal lumen of asymptomatic carriers, passing cysts in their stool (non-invasive infection). In other cases, the trophozoites invade the intestinal mucosa (intestinal disease) and, through the bloodstream, reach extraintestinal sites in other tissues such as the liver, brain, and lungs (extraintestinal disease) with consequential pathologic manifestations [4]. Abdominal pain, diarrhoea, nausea, vomiting and flatulence are the acute symptoms of amoebiasis caused by the eukaryotic parasite *E. histolytica* [4,5,6]. Other *Entamoeba* species are mostly commensals or are said to rarely infect humans [7].

Amoebic infection is one of the most prevalent parasitic diseases in India and worldwide [8]. It is significantly associated with poor sanitation and socioeconomic status than the geographical location's climate [9]. In most cases, amoebiasis is routinely diagnosed by light microscopy of a wet smear or stained stool samples [10]. This technique is inexpensive and simple, but it has limited sensitivity and specificity, such as being unable to differentiate between the cyst and trophozoites of the pathogenic species *E. histolytica* and the rest commensal species like *Entamoeba dispar* and the newly identified *Entamoeba bangladeshi* and the amphizoic amoeba *E. moshkovskii* which sporadically infects humans [3,10,11]. These four species are genetically related and morphologically indistinguishable with different biochemical features [3]. Amoebiasis affects approximately 50 million people in tropical regions and nearly 100,000 deaths are reported annually [12]. After malaria and schistosomiasis, it is the third foremost parasitic cause of death in humans [13]. Although all deaths could be due to invasive *E. histolytica* infestation, the prevalence is overestimated due to its epidemiological overlap with other morphologically indistinguishable species, specifically *Entamoeba dispar* and *Entamoeba moshkovskii* [14]. Moreover, cysts of another nonpathogenic amoeba, *Entamoeba hartmanni* can be mystified with *E. histolytica* under the microscope which is even smaller in size (> 10 μm) [15–16]. In many geographical areas, the actual prevalence of each species is not well characterized particularly for *E. moshkovskii*, as some reports suggest that it has a potential role in provoking human disease [17]. So, it is important to understand the molecular epidemiology of *E. moshkovskii* in endemic countries in the study of amoebiasis and it has become crucial in the last decade.

*E. moshkovskii* was first described as a distinct species from Moscow by Tshalaia in 1941. It was primarily considered to be a free-living environmental strain and still regarded as a common protozoan species found in anoxic sediments to brackish coastal pools. It is osmotolerant in nature and can be cultured in various media suitable for intestinal protozoa, in which it grows easily at temperatures of 10–15° C and 37° C [18–19]. Although all the characteristics differentiate *E. moshkovskii* from *E. histolytica* and *E. dispar*, the entire life-cycle, including excystment and metacystic development, closely resemble *E. histolytica* and *E. dispar*. The size of the amoeba trophozoites varies from 10 to 120μ, with an average of 25μ and cysts vary from 5 to 16 μ, with an average of 10μ [19]. In 1961, an *E. histolytica*-like strain was obtained from a resident of Laredo, Texas, who suffered from diarrhoea, weight loss, and epigastric pain and the strain was named *E. histolytica* Laredo strain which shared many biological characteristics with *E. moshkovskii*. Both the Laredo strain and *E. moshkovskii* grew easily at room temperature, were osmotolerant, and resistant to drugs used in the chemotherapy of amoebiasis, for instance, emetine [20]. Subsequent molecular studies revealed that the *E. histolytica* Laredo strain is *E. moshkovskii*, the first human isolate of *E. moshkovskii* [20–21].

Nowadays importance of *E. moshkovskii* is increasing in the study of amoebiasis, and it is reported as a common *Entamoeba* infection in humans in some settings. Colonization of *E. moshkovskii* in human hosts has been reported in countries such as the United States, Italy,

Iran, Turkey, Bangladesh, India (Pondicherry), Kenya, Australia, Indonesia, Colombia, Malaysia, Tunisia, Tanzania and Brazil [22–34]. Most of the stool samples in these studies were submitted to clinical microbiology laboratories from patients with gastrointestinal complications, indicating that *E. moshkovskii* might be associated with pathogenicity. In India (Pondicherry), it is reported that *E. moshkovskii* cannot invade intestinal mucosa and does not have any ingested erythrocytes, unlike that *E. histolytica*. [29]. In HIV-1-infected persons in northern Tanzania, *E. moshkovskii* is also not associated with clinical indicators. But in Bangladesh, this species has been identified as the only likely pathogen in individuals with gastrointestinal clinical manifestations, including dysentery [26,27]. However, in these patients, no studies of viral or bacterial agents were conducted to rule out other pathogens or potential pathogens. Another study in Bangladesh by Shimokawa et al, 2012 also pointed out a possible cause of diarrhoea in infants, which was due to *E. moshkovskii* infection [27]. While in Malaysia, it was isolated from both symptomatic and asymptomatic cases [28]. In the murine model of intestinal amebiasis, *E. moshkovskii* also caused diarrhoea, weight loss, and colitis [27]. Thus, the pathogenicity of *E. moshkovskii* in humans remains unclear.

For parasite identification, phylogeny and genetic characterization, small subunits of nuclear ribosomal RNA (18S rRNA) loci are recognized as potential targets [35–38]. Therefore, amplification and sequencing of this gene is being extensively used for decades [2,12,26,31–32]. Moreover, genetic variations based on 18S rRNA can be significant in pathogenicity for parasites.

To date, limited studies have been conducted on the genetic and phylogenetic characterization of *E. moshkovskii* in India [29] and remain largely unknown. In this study, we aimed to determine the epidemiology and molecular characterization based on 18S rRNA of amphizoic amoeba *E. moshkovskii* in human stool samples from an active surveillance study on enteric pathogens in and around Kolkata, India. We have also investigated the level of genetic diversity and established the genetic structure among the obtained local isolates using a molecular analysis tool.

## Material and methods

### Ethical statement

The ethical clearance of this study was reviewed and approved by the Institutional Human Ethics Committee of the Indian Council of Medical Research-National Institute of Cholera and Enteric Diseases (IRB Number: A-1/2015-IEC). Written informed consent statements were obtained from every participant. In the case of children, voluntary written informed consent statements were taken from their caregivers (parent/guardian).

### Study area and population

This is a hospital-based systemic surveillance study conducted from March 2017 to February 2020. The target population is patients from different parts of Kolkata and adjacent areas admitted to Infectious Disease & Beliaghata General Hospital and Dr. B C Roy Post Graduate Institute of Paediatric Sciences Hospital, Kolkata, with diarrheal complaints. These two hospitals are referral centres for the treatment of diarrheal diseases along with government health care facilities. Non-diarrhoeal cases have not been included in this study.

### Sample collection and microscopy

This is a hospital-based systemic surveillance study conducted among the patients admitted to Infectious Disease & Beliaghata General Hospital and Dr B C Roy Post Graduate Institute of

Paediatric Sciences Hospital, Kolkata, India, with diarrheal complaints from March 2017 to February 2020. These two hospitals are referral centres for the treatment of diarrheal diseases along with government health care facilities. Non-diarrhoeal cases have not been included in this study. This study collected and screened a total of 3258 samples from ID Hospital and 2793 from B C Roy Hospital. The faecal samples were collected from the patients admitted to the hospital by trained medical professionals in the presence of attending physicians on the first day of hospitalisation and before antibiotic therapy. Samples were collected in a sterile container with a unique identification number. Once the samples were collected, they were immediately sent to laboratories for testing after written informed consent was obtained from patients. All stool samples were microscopically examined in triplicate in saline and iodine wet mounts. Trichome staining also has been used for *E. moshkovskii* identification. The Uninuclear, binuclear, trinuclear, or tetranuclear cysts or trophozoites of *Entamoeba* spp observed were recorded.

## DNA extraction

DNA was isolated directly from clinical samples using STOOL DNA Minikit (QIAGEN, USA) as per the manufacturer's protocol, followed by genetic identification through PCR amplification via species-specific primers. For molecular detection of parasites, species-specific primers have been generated using Primer3 Software.

## Primer design

We have designed a nested PCR system targeting the 18S rRNA gene to detect *E. moshkovskii* from stool samples. The first set of primer (outer primer set) was used to amplify the 18S rRNA locus of the *Entamoeba* genus (Genus specific PCR assay), and the second set of primer (Nested primer set) was used for the identification of *E. moshkovskii* species only. The forward and reverse primers of the *Entamoeba* genus were designed from highly conserved regions of 18S rRNA locus of five phylogenetically close *Entamoeba* species viz. *E. histolytica*, *E. dispar*, *E. moshkovskii*, *E. bangladeshi* and *E. nuttalli*. The following sequences were analysed: *E. histolytica*, GenBank accession no. X56991 (1947 bp); *Entamoeba dispar*, GenBank accession no. Z49256 (1949 bp); *E. moshkovskii*, GenBank accession no. AF149906 (1944 bp); *E. bangladeshi*, GenBank accession no. KR025411 (1927 bp); *E. nuttalli*, GenBank accession no. AB485592.1 (2431 bp); *Entamoeba coli*, GenBank accession no. AF149914 (2101 bp); *Entamoeba chattoni*, GenBank accession no. AF149912 (1963 bp); *Entamoeba polecki*, GenBank accession no. AF149913 (1858 bp); *Entamoeba invadens*, GenBank accession no. AF149905 (1965 bp). The sequences were aligned using the ClustalW program (https://www.genome.jp/tools-bin/clustalw). Primer 3 online software was used to design the primers. Primer length and melting temperature were considered. Primer sequences specific for *Entamoeba* spp. were as follows: EntaS_F: CTGCCAGTATTATATGCTGATGTT and EntaS_R: TCTCCTTCCTCTAAATA AGGAGATTTA. The ability of genus-specific primer to amplify the 18S rRNA of the five *Entamoeba* species was verified using genomic DNA preparation. In the second round, a nested primer set was designed to amplify the *E. moshkovskii* species DNA fragment of the same gene. The nested *E. moshkovskii*-specific primer sequences were EM_779bpNF: AACTAACGAAG GAGATGAAGTGAG and EM_779bpNR: GCCAGAGACATCGATTAAAATG.

## PCR amplification

The genus-specific PCR assay was adjusted to ensure the amplification of each target. We performed PCR amplification in a final volume of 50 mL containing 1X PCR buffer and 1U of Biotaq DNA polymerase (Bioline, UK) to obtain the primary PCR product. After $MgCl_2$

concentrations ranging from 1.0–4.0 mM were checked, the optimal concentration of $MgCl_2$ was found to be 2.5 mM. 0.2 μM of each forward and reverse primer was found to be the optimal concentration for best results. After running a gradient PCR, the optimal annealing temperature was fixed to 55˚C. The amplification was done in a thermal cycler as follows: 5 min at 94˚C, followed by 35 cycles, each of 94˚C for 45 sec, 55˚C for 50 seconds and then 72˚C for 50 seconds with a final extension at 72˚C for 7 mins. Amplified PCR products were separated by agarose gel electrophoresis and visualised in a UV transilluminator after 0.5 μm/ml of ethidium bromide staining. The genus-targeted PCR products of *Entamoeba* spp showed 1803 bp—2046 bp amplicon on the agarose gel, depending on the species.

For the detection of *E. moshkovskii* the primary products went through the second round of PCR amplification using species-specific primer pair. 1.0 μL of primary PCR product was used as a template for the nested PCR reaction. To obtain the nested PCR product, we performed PCR amplification in a final volume of 50 μL reaction mixture containing 1X PCR buffer, 4.0 mM $MgCl_2$, 0.2 mM of each dNTP, 0.2 μM of each forward and reverse primer (EM_779bpNF & EM_779bpNR) and 1U of Biotaq DNA polymerase (Bioline, UK). Reactions were performed in a thermal cycler PCR system (Applied Biosystem). The PCR reaction was started with an initial denaturation step at 94˚C for 3 minutes and then subjected to 35 cycles of 94˚C for 40 seconds, 57˚C for 35 seconds and 72˚C for 45 Seconds, followed by a final extension at 72˚C for 7 minutes. The nested PCR was performed in 2.0 mM $MgCl_2$. Nested PCR generates a 779 bp fragment in the presence of *E. moshkovskii*. Amplified PCR products were separated by electrophoresis in 1.5% agarose gel (Lonza SeaKem® LE Agarose)s in 1X Tris boric Acid EDTA buffer and visualised in a UV transilluminator after 0.5 μm/ml of ethidium bromide staining.

The specificity of Genus specific PCR was tested using genomic DNA preparation of five *Entamoeba* species viz. *E. histolytica*, *E. moshkovskii*, *E. dispar*, *E. bangladeshi and E. nuttalli*. The specificity of *E. moshkovskii* species-specific primer set was also assessed against DNA extracted from faecal samples of other pathogens, namely *E. histolytica*, *E. dispar*, *E. bangladeshi*, *E. nuttalli*, *E. coli*, *Giardia lamblia*, *Cryptosporidium parvum*, *Cryptosporidium hominis*, *Cryptosporidium viatorum* and mixed bacterial infections. All the tested DNA samples were subjected to the abovementioned amplification protocol. The sensitivity of the nested PCR system was also evaluated using reference DNA templates by serial dilutions from 10 to 0.000019 ng/μL of DNA.

## DNA Sequencing

The 18S rRNA locus of the positive samples was sequenced to characterize the local isolates of *E. moshkovskii.* As the same-size amplified products do not unavoidably mean identical DNA sequences, we directly sequenced the PCR products without cloning them into any vector to reduce the chances of any sequence selection. For DNA sequencing of *E. moshkovskii* positive samples obtained in this study were amplified separately using ExTaq DNA polymerase (Takara, Japan). We used ExTaq because of its higher fidelity than standard Taq with a lower mutation rate. The aforementioned species-specific primer set was employed directly to amplify the 18S rRNA locus of the positive samples by conventional PCR method. PCR was accomplished in a 50 μL reaction mixture containing: 0.50 μL Takara Ex Taq, 5 μL 10X Ex Taq Buffer, dNTP mixture 3 μL, 0.2 μM of each forward and reverse primer (EM_779bpNF & EM_779bpNR), approximately 200 ng stool DNA and nuclease-free water (Ambion™) up to 50 μL. The reaction mixture was subjected to an initial denaturation step at 94˚C for 5 mins, followed by 35 cycles of 60 s at 94˚C of denaturation, primer annealing for 40 s at 57˚C and extension for 55 s at 72˚C. A final seven minutes polymerization step at 72˚C was also

performed. Successfully amplified PCR products were purified using the Roche PCR Gel extraction Kit as per the manufacturer's protocol. The purified PCR products were sequenced using the standard BigDye terminator V3.1 sequencing kit (Applied Biosystem, USA) following the manufacturer's instructions. Sequencing was performed with a 5730 DNA analyzer (Applied Biosystem, Foster City, CA, USA). The accuracy of the sequence was verified with sequencing in the 5' - 3' direction using both forward and reverse primer separately.

## Sequence alignment, nucleotide polymorphisms analysis

The obtained 18S rDNA gene sequences of *E. moshkovskii* compared to those available in the GenBank database using the BLAST tool (NCBI - https://blast.ncbi.nlm.nih.gov/Blast.cgi)). All of the obtained sequences were deposited in NCBI GenBank (accession numbers ON965383-ON965450). Multiple alignments of the nucleotide sequence allowed us to analyze nucleotide sequence variations. The obtained sequences were aligned using ClustalW multiple sequence alignment program of GenomeNet Bioinformatics tools and edited manually. This alignment was also performed with MultAlin using identity parameter values of -1 and -0 and the penalty default values to determine sequence variations. All obtained sequences were aligned and adjusted in MEGA X. Substitution matrix (Maximum likelihood/ML) and transition/transversion (ML) bias were estimated using the same software. In the substitution matrix (ML), each entry is the probability of substitution ($r$) from one base (row) to another base (column). Substitution patterns and rates were estimated under the General Time Reversible model (+G+I) [39]. A discrete Gamma distribution was used to model evolutionary rate differences among sites (5 categories, [+*G*], parameter = 200.0000). The rate variation model allowed some sites to be evolutionarily invariable ([+*I*], 0% sites). Rates of different transitional substitutions are shown in bold and those of transversionsal substitutions are shown in italics. Relative values of instantaneous *r* should be considered when evaluating them. For simplicity, the sum of *r* values is made equal to 100. For estimating Maximum likelihood values, a tree topology was automatically computed [39]. This analysis involved 68 nucleotide sequences. For transition/transversion (ML) bias determination substitution patterns and rates were estimated under the Tamura-Nei (1993) model (+G+I) [40]. There were a total of 733 positions in the final dataset. Nucleotide composition, parsimony-informative sites, singleton sites, variable sites (S), the number of haplotypes (h), haplotype diversity (Hd), the average number of nucleotide differences (K), and nucleotide diversity ($\pi$) were determined from the aligned sequences using the program DnaSP v5.

## Genetic structure analysis

Partitions of genetic diversity within and among different population subdivisions of local isolates obtained from the present study were calculated using Wright's F statistics ($F_{ST}$). Population subdivision was performed based on the co-infection status of the isolates. $F_{ST}$ is a measure of genetic differentiation among populations. DnaSP v5.0 package was used to estimate the mean pairwise differences between the populations. The significant difference in $F_{ST}$ was from 0 based on 1000 random permutations of the dataset. Four neutrality tests using Tajima's D, Fu's FS, Fu and Li's D and F statistics were also performed through DnaSP v5.0 for assessing the probable population expansion.

## Phylogenetic analysis

The maximum Likelihood method and Tamura-Nei model were applied to infer the evolutionary history [40]. The tree with the highest log likelihood (-6543.31) is shown. Initial tree(s) for the heuristic search were obtained automatically by applying Neighbor-Join and BioNJ

algorithms to a matrix of pairwise distances estimated using the Tamura-Nei model and then selecting the topology with superior log likelihood value. A discrete Gamma distribution was used to model evolutionary rate differences among sites [8 categories (+$G$, parameter = 200.0000)]. The rate variation model allowed for some sites to be evolutionarily invariable ([+$I$], 0.42% sites). The tree is drawn to scale, with branch lengths measured in the number of substitutions per site. This analysis involved 76 nucleotide sequences. There were a total of 822 positions in the final dataset. Evolutionary analyses were conducted in MEGA X [39,41].

## Haplotype network construction

The relationship between the haplotypes of *E. moshkovskii* in different coinfected subgroups was inferred by constructing a Median Joining haplotype network in PopART v1.7. The isolates were colour-coded according to specific coinfection groups to derive a relationship with sequenced data.

## Data collection of other Enteropathogen infections

*E. histolytica*, *Cryptosporidium* and *Giardia* were identified by PCR method as described elsewhere [42]. Helminth parasites were detected by light microscopy after wet mount. Other enteric pathogens like *Vibrio cholera* O1/O139, *Salmonella* spp., *Campylobacter jejuni*, Rotavirus, astrovirus and adenovirus co-infection information with *E. moshkovskii* were obtained from the institutional database of ICMR-National Institute of Cholera and Enteric Diseases, Kolkata, India.

## Cultivation of *Entamoeba* spp

In this study, we used Trophozoites of the *E. moshkovskii* (An isolate of Thailand), a gift from Dr Seiki Kobayashi, School of Medicine, Keio University. The *E. moshkovskii* strain was regularly cultivated in our lab using the BIS-33 medium and maintained at a temperature of 25˚C. Trophozoites under the log phase of growth were used in the experiments as a positive control.

## Geographical distribution of *E. moshkovskii* infection

The geographical locations of the patients were obtained from the hospital case record file (CRF). This data was employed to evaluate the spatial distribution of *E. moshkovskii* in the catchment area. If patients from any area comprised less than 1% of the total sample size, the site was excluded from the dataset.

## Statistical analysis

GraphPad prism v.8.4.2, CA, USA, was used to analyze the data. The relationship between the prevalence of *E. moshkovskii* with other variables like age, gender, and other co-infections/ additional enteropathogens was measured by testing $X^2$. Fisher's exact test was applied to evaluate the statistical association between gender and the occurrence of infection of *E. moshkovskii*. Two-way ANOVA was employed in three-year sample comparisons to assess the statistical significance of the seasonal pattern of *E. moshkovskii* infection. In all cases, a p-value less than 0.05 was considered significant.

## Results

### Prevalence of *Entamoeba moshkovskii*

A total of 6051 samples (3258 samples from ID Hospital and 2793 samples from B C Roy Hospital) were collected and screened in this study to detect *E. moshkovskii*. Out of 6051 clinical samples, 4.10% (n/N = 248/6051, 95% CI 0.036–0.046) samples were identified as cysts or trophozoites of *Entamoeba* spp. by light microscope (Fig 1). Genus-specific PCR assay showed 4.84% (n/N = 293/6051, 95% CI 0.043–0.054) samples were positive for *Entamoeba spp*. As a result, microscopy failed to detect 15.36% (n/N = 45/293, 95% CI 0.273–0.380) of the samples that were positive for *Entamoeba* spp. All the samples that tested positive in the genus-specific PCR assay (including those that were positive and negative for *Entamoeba* spp by microscopy) were then subjected to a second round of nested PCR to screen for *E. moshkovskii* infection. The prevalence of *E. moshkovskii* after nested PCR assay was 3.12% (n = 189/6051, 95% CI 0.027–0.036). Therefore, 1.72% (n = 104/6051, 95% CI 0.014–0.021) of patients were infected with other *Entamoeba* species in the study area. 64.50% (n/N = 189/293, 95% CI 0.59–0.70) of amoebic infections were positive for *E. moshkovskii* in this study area. The PCR system utilized in the prevalence study possessed the capability to identify as little as 1.2 pg genomic DNA of *E. moshkovskii*. These results increased the attention to studying traditionally non-pathogenic species infection and invasive amoebiasis. Approximately 19.04% (n/N = 36/189, 95% CI 0.141–0.252) of *E. moshkovskii* infections were overlooked using microscopy but were later identified using a nested PCR test. The finding suggests that the number of cysts produced in *E. moshkovskii* infections is smaller when compared to other *Entamoeba* species. Of the 189 samples that tested positive for *E. moshkovskii* through PCR, only 9.00% (n/N = 17/189, 95% CI 0.056–0.14) showed the presence of trophozoites through microscopy.

The occurrence of *E. moshkovskii* was higher in Males (*E. moshkovskii*: n/N = 117/189, 61.35%, 95% CI 0.548–0.685) than in females (*E. moshkovskii*: n/N = 38/189, 38.62%, 95% CI 0.315–0.452); but the difference between them was not statistically significant (Fisher's exact test = 0.2548, P<0.05). The infections (Sole) were significantly associated with age groups ($X^2$ = 26.01, P<0.0001). The highest (n/N = 54/189, 28.57%, 95% CI 0.223–0.354) and lowest (n/N = 12/189, 6.35%, 95% CI 0.036–0.109) infection rate of *E. moshkovskii* was found among 5–12 and 19–29 years of age groups, respectively (Fig 2A).

### Additional enteropathogens in stool samples of *E. moshkovskii* (sole) infected peoples

Many bacteria, viruses, enteric parasites, and helminths are associated to induce diarrhoea. The coinfection status with other diarrheagenic organisms was analyzed by examining the data from our institutional database, followed by evaluating the statistical significance. In the 189 diarrheal stool samples with *E. moshkovskii*, few were coinfected with one kind of enteropathogen, like 5.82% (n/N = 11/189, 95% CI 0.032–0.102) with *E. histolytica*, 2.12% (n/N = 4/189, 95% CI 0.008–0.053) with *G. lamblia*, 3.17% (n/N = 6/189, 95% CI 0.015–0.068) with *Cryptosporidium* spp, 7.94% (n/N = 15/189, 95% CI 0.049–0.127) with Soil-transmitted helminths (STH), 12.70% (n/N = 24/189, 95% CI 0.087–0.182) with *E. coli*, 10.58% (n/N = 20/189, 95% CI 0.007–0.158) with *Shigella* spp, 4.23% (n/N = 8/189, 95% CI 0.022–0.081) with *V. cholerae*, 2.12% (n/N = 4/189, 95% CI 0.008 0.053) with Rotavirus (Fig 2B). 4.76% (n/N = 9/189, 95% 0.025–0.088) samples were co-infected with 2 or >2 enteropathogens. 46.56% (n/N = 88/189, 95% CI 0.396–0.537) of the samples were positive solely for *E. moshkovskii* (Fig 2B). So it is, therefore, worthy of note that the diarrheal cases associated with *E. moshkovskii* were not commonly coinfected in Kolkata. Many positive samples were infected with *E. moshkovskii*,

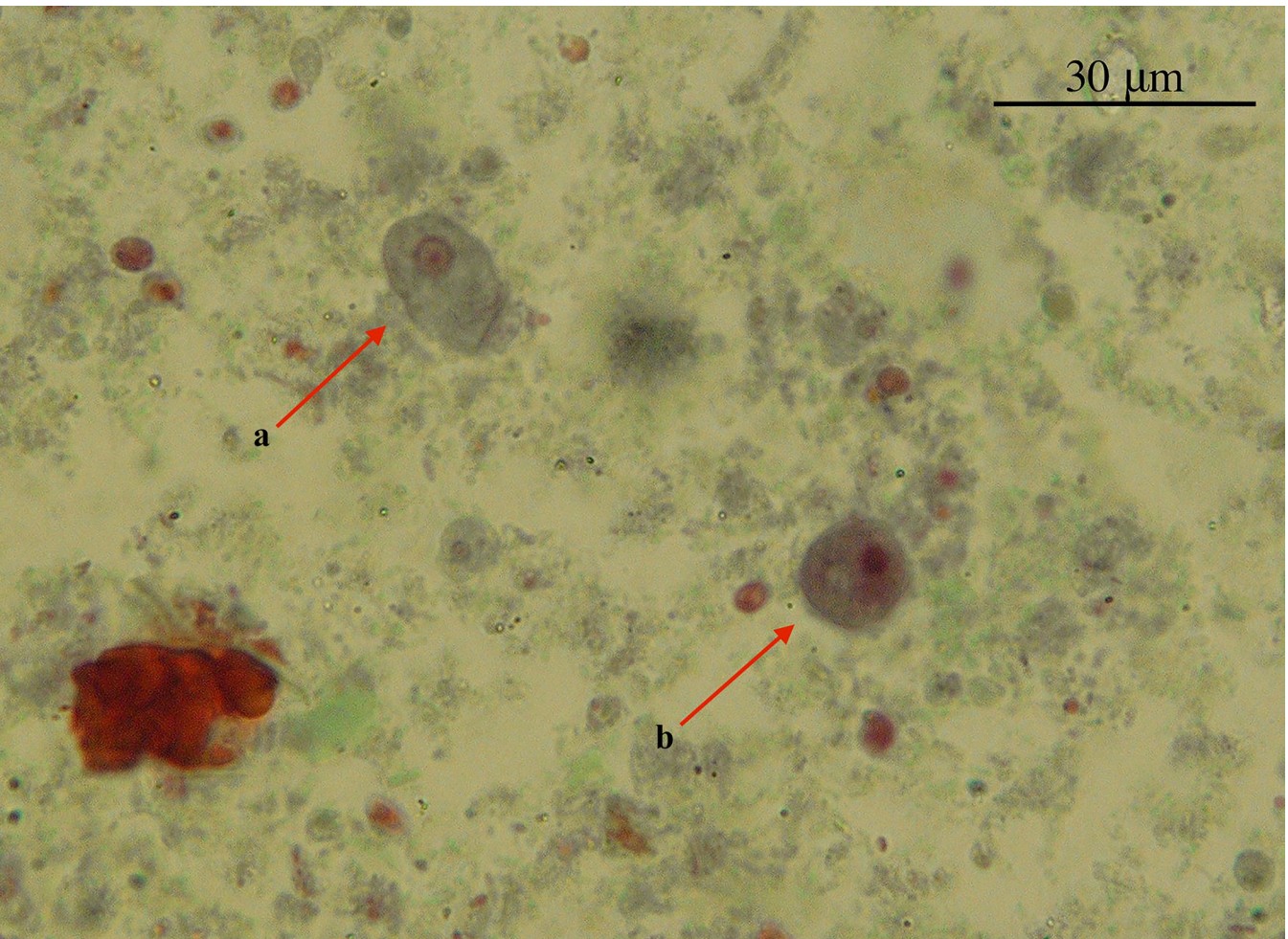

**Fig 1. Microscopic view of *Entamoeba moshovskii* cysts and trophozoites after Trichrome staining (40X).** a. Trophozoite of *Entamoeba moshkovskii*.; b. Cyst of *Entamoeba moshkovskii*.

and the sole infection with *E. moshkovskii* was significantly associated with diarrheal incidence ($X^2$ = 335.5, df = 9; P<0.0001) in this study area.

### Seasonal distribution of *Entamoeba moshkovskii* infection

In this three-year study (March 2017 to February 2020), a specific seasonal distribution was found for *E. moshkovskii* infection. It was observed that the prevalence of *E. moshkovskii* parasites mostly increased during the post-fall (Sep-Nov) season and in the summer season (April-June) of each year and a statistically significant correlation was observed (p<0.0001) (Fig 2C). Moreover, the prevalence of *E. moshkovskii* was significantly higher than that *E. histolytica* throughout the three years (p = 0.0095).

### *E. moshkovskii* prevalent areas

The prevalence study carried out in the I.D. hospital and B C Roy Children Hospital pointed out that the following areas i.e., Beliaghata, Entally, Kashipur-Belgachia, Maniktala, Gopalpur-Rajarhat, Kolkata-Port, Jorasanko, Baruipurpurba, Maheshtala, Metiabruz, Bhangar around Kolkata were very highly burdened (>3.5%) with *E. moshkovskii* infection. Ballygunge,

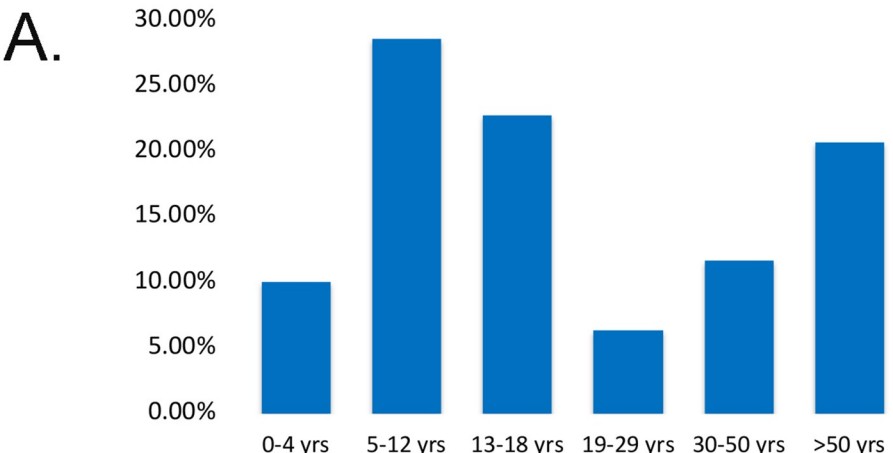

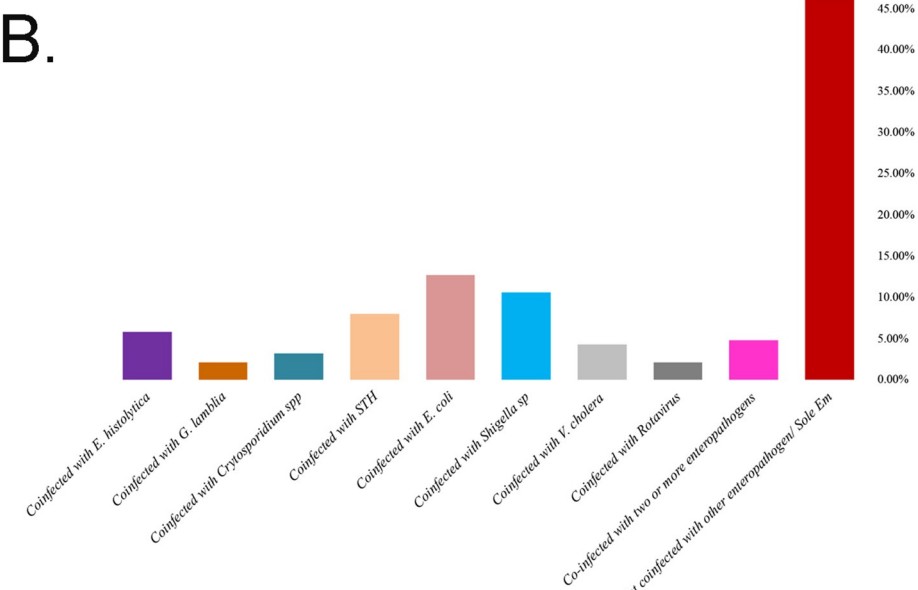

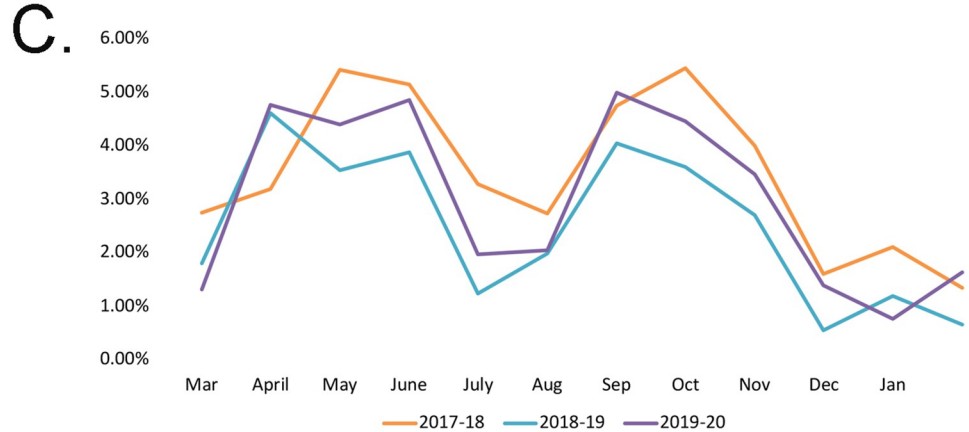

**Fig 2.** A. Age-wise distribution of *Entamoeba moshkovskii* (mono-infection); Total number of infected individuals = 189. B. Additional enteropathogens in *E. moshkovskii* infected patients. C. Seasonal distribution of *Entamoeba moshkovskii* from March 2017 to February 2020 in Kolkata and adjacent areas.

Chowringhee, Dumdum Uttar, Dumdum, Bidhannagar, Bhabanipur and Kashba had high prevalence rates (3%-3.5%). Behala Paschim, Behala Purba, and Jadavpur were moderately (2.5%-2.99%) infected, while the lowest prevalence ($< 2.49$%) was reported in Baranagar, Rajarhat New Town and Shyampukur areas.

## Phylogenetic and sequencing analysis of local isolates of *E. moshkovskii*

To consider the genetic diversity in *E. moshkovskii* isolates in and around Kolkata, the 18S rRNA gene locus in 68 positive cases was successfully sequenced. Sequencing revealed that the locally found *E. moshkovskii* strains were 99.59% - 100% identical with the prototype (GenBank accession no KP722605.1), and all isolates were the same species. All 68 sequences were deposited in NCBI GenBank under accession numbers ON965383- ON965450 [S1 Table]. The nucleotide sequence of *E. moshkovskii* of 36 isolates had 100% genetic identity with the reference sequences (prototype) previously published in GenBank (accession numberAF149906.1), and the rest of the 32 sequences represented genetic variants of *E. moshkovskii* not described earlier. We selected five known sequences, viz. *E. histolytica* (Genbank: AP023147.1), *E. nuttalli* (Genbank: LC042219.2), *E. invadens* (Genbank: AF149905.1), *E. chattoni* (Genbank: AF1499121.1), *E. coli* (Genbank: AF149914.1), *Dictyostelium discoideum* (Genbank: AM168039.1), *E. polecki* (Genbank: AF14913.1) to construct the phylogenetic tree of the *E. moshkovskii* isolates obtained in this study. Phylogenetic analysis showed that the study isolates of *E. moshkovskii* formed a monophyletic sister clade to the common ancestors of the *E. histolytica* and *E. dispar* 18S rRNA sequences (Fig 3). The clades were supported with high bootstrap values. The studied *E. moshkovskii* isolates were grouped with both prototypes (GenBank accession no: AF149906.1) and their close variants. Most of the variants showing the bootstrap value of 89%–99% shared a common clade with the prototype cluster, consisting of two subgroups: sole infection and mixed infection. Some isolates from the mixed infection subgroup are also clustered into separate distant clades from the prototype lineage. The EM IND/37 (GenBank: ON9654419) and EM IND/46 (GenBank: ON9654428) formed a separate group with 100% bootstrap value from the prototypes, which was considered unique due to the presence of one insertion (nucleotide A) and one deletion (nucleotide T) at the studied locus of the isolates. This group requires further investigation to gain a better understanding of the evolutionary processes that have shaped them.

## Overall distribution of SNPs in 18S rRNA locus of *E. moshkovskii*

The sequence of 18r RNA locus from 68 isolates was successfully acquired from positive stool samples. This analysis involved all of the 68 nucleotide sequences. A total of 733 positions were analysed to identify SNPs in the final dataset. Through the sequence of 18S rRNA locus, we have identified 10 SNPs, one deletion and one insertion in 733 sequenced bases from 68 isolates of *E. moshkovskii* (Table 1). The isolates were 99.60%- 100% identical (GenBank accession no AF149906.1), and all isolates were of the same species (S1 Table). Out of 68 isolates, 36 had sequences identical to the corresponding reference sequence and the rest of the 30 differed by one to three single-nucleotide polymorphisms (SNPs). Two samples had one insertion at position 795–796 (nucleotide A) and one deletion at position 769 (Nucleotide T). All SNPs identified corresponded to transition (pyrimidine $< = >$ pyrimidine/ purine $< = >$ purine) or transversion (purine $< = >$ pyrimidine) mutations. The studied locus of the isolates transition

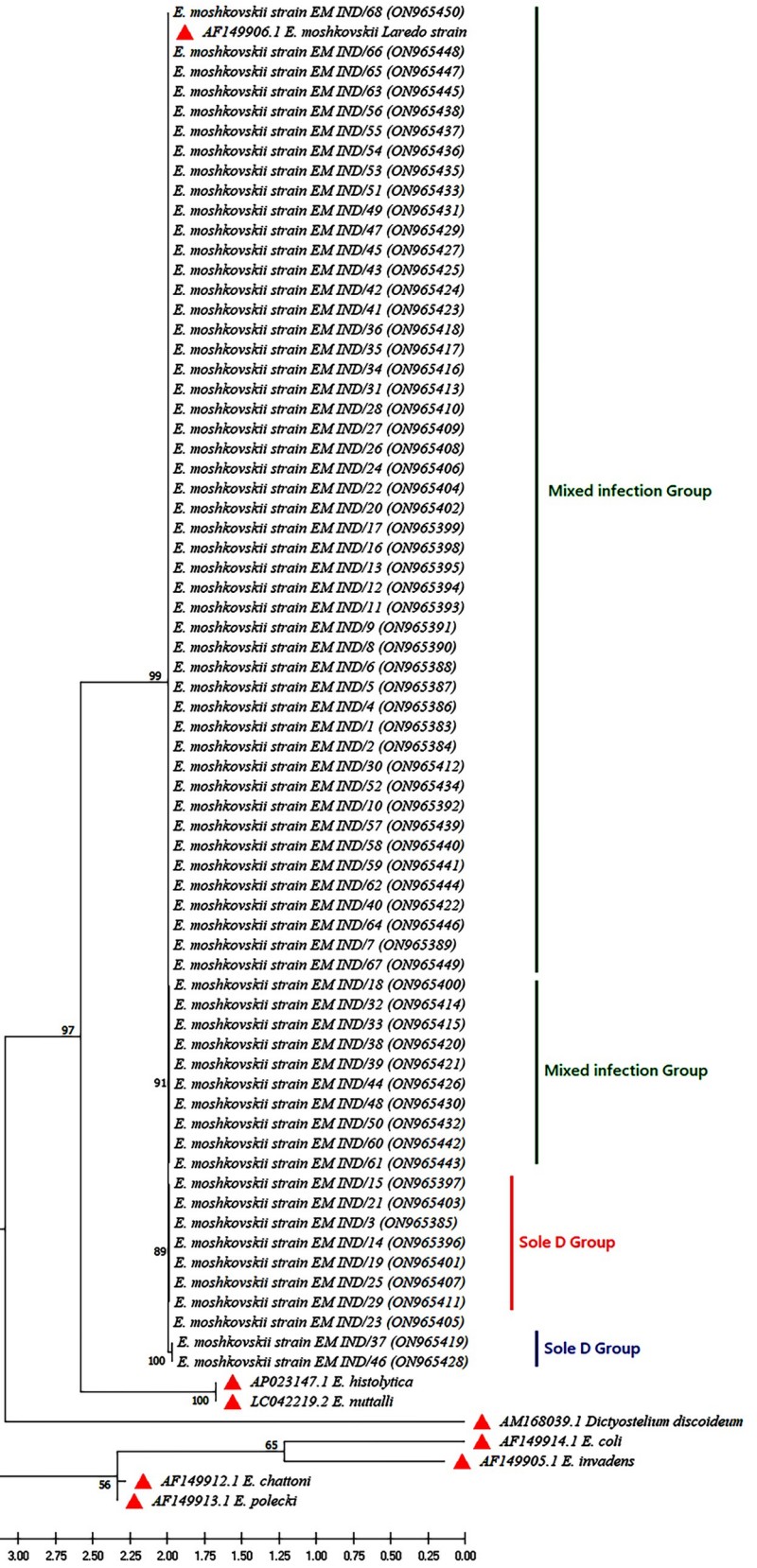

**Fig 3. Phylogenetic analysis of study isolates based on 18S rRNA locus of *Entamoeba moshkovskii*.**

**Table 1.  SNPs/ insertion-deletion identified in 18S rRNA of *E. moshkovskii* study isolates.**

| SNPs | P VALUE | X², df | Significance | Associated group |
|---|---|---|---|---|
| 722 T/C | <0.0001 | 34.24, 4 | Yes | IEH |
| 788 T/C | 0.864 | 1.286, 4 | No | _ |
| 814 T/G | 0.0142 | 12.46, 4 | YES | IBV |
| 826 T/A | 0.0142 | 12.46, 4 | YES | IBV |
| 988 G/C | 0.293 | 4.945, 4 | No | _ |
| 1145G/A | 0.1909 | 6.113, 4 | No | _ |
| 1345 T/G | 0.0424 | 9.884, 4 | Yes | Sole D |
| 1361 A/G | 0.0424 | 9.884, 4 | Yes | Sole D |
| 1377 T/G | 0.1448 | 6.836, 4 | No | _ |
| 1437 G/A | 0.2445 | 5.446, 4 | No | _ |
| 769 A delete | 0.3295 | 4.612, 4 | No | _ |
| 795–796 T insert | 0.3295 | 4.612, 4 | No | _ |

Sole D: diarrheal patients solely infected with *E. moshkovskii*, IEH: *E. moshkovskii* positive samples co-infected with *Entamoeba histolytica*, IOEP: *E. moshkovskii* positive samples co-infected with other Enteric Parasites- *G. lamblia*, *Cryptosporidium* spp, ISTH: *E. moshkovskii* positive samples co-infected with soil-transmitted helminths, IB/V: *E. moshkovskii* positive samples co-infected with other diarrhoea-causing bacteria-*E. coli*, *Shigella* spp & *V. cholera* or virus-Rotavirus., P: Correlation coefficient value of the particular association, df: degree of freedom, X²: Chi-square value.

and transversion base substitutions showed the same propensity (50%). The estimated Transition/Transversion bias (*R*) was 0.4. The average base composition in the studied locus was A = 33.57%, T/U = 25.30%, C = 16.80%, and G = 24.33%, with slightly high AT richness in the sequences. The maximum Log-likelihood for this computation was -1181.283. The maximum likelihood estimate of the substitution matrix is presented in S2 Table. We have also obtained a sum of 28 polymorphic/variable sites, out of which there were 0 singleton variables and 28 were parsimony-informative sites (Table 2). The parsimony-informative sites gave information about the evolutionary relationships among the isolates. Among 10 SNPs observed in the

**Table 2.  Genetic diversity indices and neutrality tests based on 18S rRNA sequences found in local isolates of *E. moshkovskii* in Kolkata and adjacent areas.**  Haplotype_1(Prototypes) was identical to Laredo strain of *E. moshkovskii*.

| Subgroups | N | Haplotypes obtained | S | h | K | Hd +/-SD | Π +/-SD | Fu's *Fs* | Tajima's D | Fu & Li's D | Fu & Li's F |
|---|---|---|---|---|---|---|---|---|---|---|---|
| Sole D | 30 | Hap_1, Hap_2, Hap_3, Hap_4, Hap_5, Hap_6, Hap_7, Hap_8 | 9 | 8 | 1.42759 | 0.623 +/-0.093 | 0.00195 +/0.00036 | -2.506 | -1.15479 | -2.12379 | -2.13591 |
| IEH | 6 | Hap_4, Hap_6 | 3 | 2 | 1.60000 | 0.533 +/-0.172 | 0.00218 +/0.00070 | 2.506 | 1.12414 | 1.39584 | 1.40624 |
| IOEP | 6 | Hap_1, Hap_3, Hap_10 | 2 | 3 | 0.66667 | 0.600 +/-0.215 | 0.00091 +/0.00038 | -0.858 | -1.13197 | -1.15529 | -1.19511 |
| ISTH | 5 | Hap_1, Hap_8 | 1 | 2 | 0.40000 | 0.400 +/-0.237 | 0.00055 +/0.00032 | 0.90 | -0.81650 | -0.81650 | -0.77152 |
| IB/V | 21 | Hap_1, Hap_6, Hap_9, Hap_10 | 22 | 4 | 4.4667 | 0.633 +/-0.074 | 0.00609 +/0.00266 | 5.167 | -1.02116 | 1.35707 | 0.75547 |
| Total | 68 | | 28 | 10 | 2.46313 | 0.677 +/-0.057 | 0.00336 +/0.00100 | -0.544 | -1.83320** | 1.85839* | 0.58884 |

N: Sample size; S: number of polymorphic/segregating sites; : number of haplotypes; K: Average number of nucleotide differences; Hd: haplotype diversity;: π nucleotide diversity

**Statistically significant, p<0.05

*Statistically significant, p<0.02

18S rRNA locus, five SNPs were potentially associated with specific coinfection incidence. For example, 722 T/C (p = <0.0001) transition was found to be associated with *E. histolytica* co-infection (IEH). 814 T/G transversion (p = 0.0142) and 826 T/A transversion (p = 0.0142) also exhibited a strong association with diarrhoea-causing bacterial or viral co-infection (IB/V). Only 1345 T/G (p = 0.0424) and 1361 A/G (p = 0.0424) showed a positive correlation with the sole infection of *E. moshkovskii*. However, the deletion of nucleotide A (adenine) at 769 and insertion of nucleotide T (thymine) at 795–796 were not significantly associated with any co-infection incidence. Detailed information on SNPs identified within the target loci of study isolates is provided in S1 Table.

## Median-Joining haplotype network

In total, 68 sequences of the 18S rRNA locus were used to assess the relationship of *E. moshkovskii* haplotypes among different coinfected subgroups. Ten distinct haplotypes were identified in this study. Haplotype 1 was widespread, occurring in both coinfected subgroups viz. IOEP, ISTH and IB/V and the sole *E. moshkovskii*-infected subgroup (Sole D). The stellate shape of the constructed network suggests a rapid expansion of the population of *E. moshkovskii* in the study region (Fig 4).

## Population structure

**Genetic diversity.** We have investigated the following descriptive statistics of genetic diversity using the software DnaSP v5: number of segregating sites (S), number of haplotypes (), haplotype diversity (HD), and nucleotide diversity ($\pi$) and Average number of nucleotide (K). Genetic diversity indices revealed a total of 28 segregating sites (S) and 10 haplotypes, as well as haplotype diversity (HD) of 0.677 ±0.057 and nucleotide diversity ($\pi$) of 0.00336 ±0.00100. We also investigated the same for different co-infected/sole *E. moshkovskii*-infected subgroups. The values are provided in Table 2. The haplotype diversity (HD) for different subgroups ranges from 0.400 ±0.237 to 0.633 ±0.074, and the nucleotide diversity ($\pi$) ranges from 0.00055 ±0.00032 to 0.00609 ±0.00266 across all sites. The average number of Nucleotide differences (*k*) for each subgroup ranged from 0.40000 to 4.4667. The study found a high degree of haplotype diversity in the Sole D subgroup and low nucleotide diversity, except for those co-infected with Soil-transmitted helminths (ISTH) which showed lower haplotype diversity. The Sole D subgroup had the most haplotypes (8 haplotypes), with a high HD value. Hence, the apparent observation from this study is that there is a high level of genetic diversity in the Sole D subgroup compared to other subgroups. The Sole D subgroup had two separate subclades in the phylogenetic tree, the first containing Haplotype_2 and Haplotype_5 with a bootstrap value of 91, and the second containing only Haplotype_9. But no clustering was seen in the phylogenetic tree for other co-infected subgroups, and there was no geographical or seasonal pattern. Most novel haplotypes were found to co-occur with the most prevalent haplotype. Haplotypes Hap_2, Hap_7, and Hap_5 were only found in the Sole *E. moshkovskii* infected subgroup, while Hap_9 appeared exclusively in the subgroup co-infected with IB/V.

Three neutrality tests (Tajima' D, Fu & Li's D and Fu & Li's F value) showed negative non-significant values for Sole D, ISTH and IOEP subpopulations (Table 2). These analyses indicated deviations from neutrality and implied population expansion (e.g., after a bottleneck event or a selective sweep) and/or purifying selection in the three infected subgroups of *E. moshkovskii*. The negative Tajima' D value of the IB/V subgroup (Tajima' D = -1.02116) also suggested that this population deviated from the standard neutral. The three neutrality tests for the IEH subgroup resulted in positive values spectacled balancing selection or sudden population contraction. However, statistically non-significant value revealed a weak selection

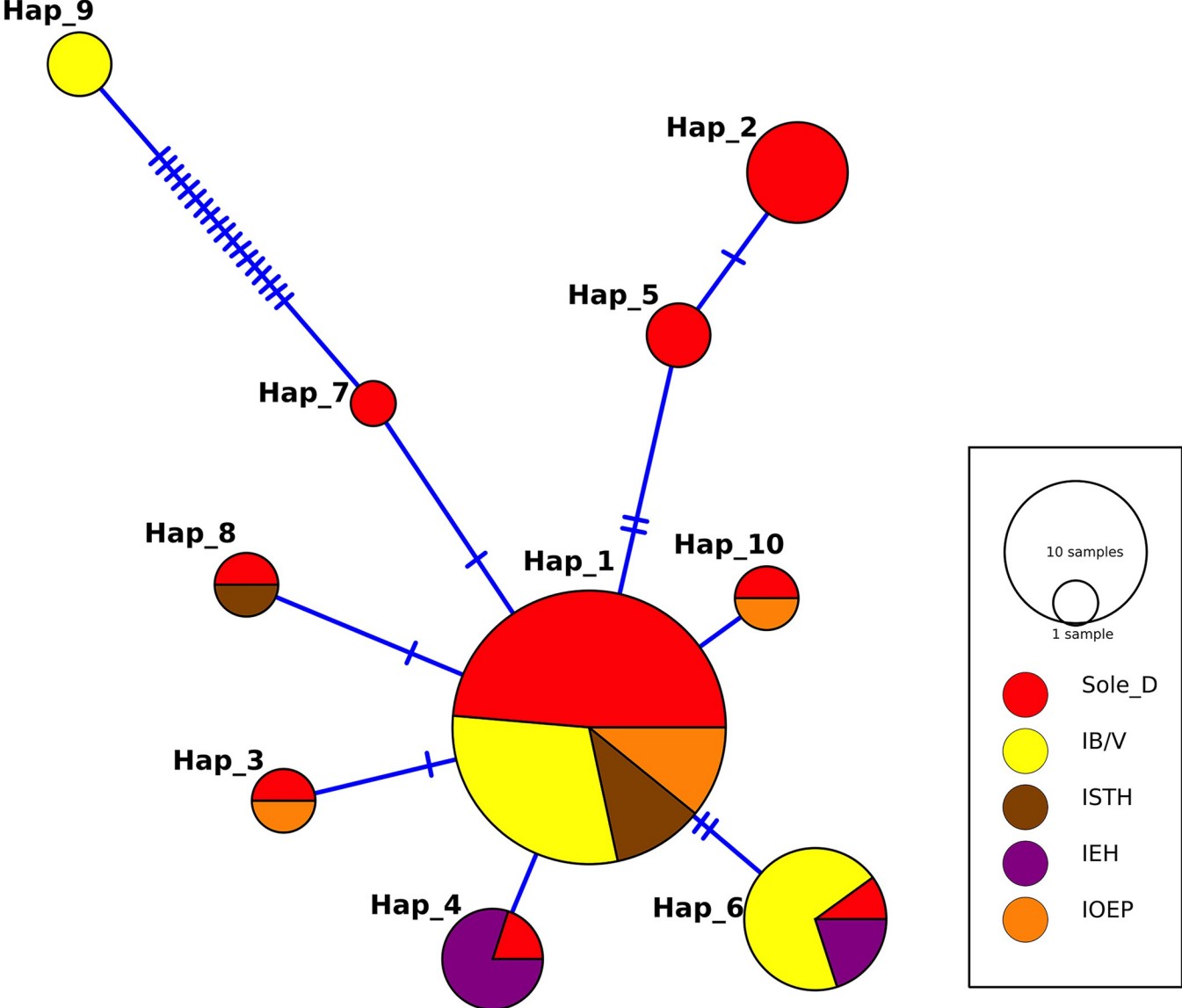

**Fig 4. Median-joining network of 68 *E. moshkovskii* haplotypes obtained in and around Kolkata, West Bengal, India.** Sole D: diarrheal patients solely infected with *E. moshkovskii*, IEH: *E. moshkovskii* positive samples co-infected with *E. histolytica*, IOEP: *E. moshkovskii* positive samples co-infected with other Enteric Parasites- *G. lamblia*, *Cryptosporidium* spp, ISTH: *E. moshkovskii* positive samples co-infected with soil-transmitted helminths, IB/V: *E. moshkovskii* positive samples co-infected with other diarrhoea-causing bacteria-*E. coli*, *Shigella* spp &*V*. *cholera* or virus-Rotavirus.

within and among all study sites. The estimation of another neutrality test, Fu's $F_S$ based on the haplotype distribution, showed negative values for Sole D and IOEP subpopulations, demonstrating an excess of rare haplotypes compared to what would be expected under neutrality. Sole D exhibited the lowest Fu's $F_S$ value (Fu's Fs = -2.506) which conferred the hypothesis of past population expansion incidents for the subgroup. However, positive values of Fu's $F_S$ in IEH, ISTH and IB/V subgroups suggested population subdivisions and the overdominant selection or bottlenecks. IB/V subgroups showed the highest degree of positive Fu's $F_S$ value (Fu's Fs = 5.167) due to the presence of haplotype (Hap_9) with insertion–deletion polymorphisms. Strong positive neutrality results for IEH (Fu's Fs = 2.506; Tajima' D = 1.12414; Fu &

**Table 3. Genetic differentiation (F_{ST}) among different coinfected subgroups of *Entamoeba moshkovskii*.**

| Sub group | Sole D | IEH | IOEP | ISTH | IB/V |
|---|---|---|---|---|---|
| Sole D | 0.0000 | | | | |
| IEH | 0.27144 | 0.0000 | | | |
| IOEP | 0.09383 | 0.32000 | 0.0000 | | |
| ISTH | 0.10412 | 0.34783 | 0.0000 | 0.0000 | |
| IB/V | 0.10865 | 0.11116 | 0.09665 | 0.10664 | 0.0000 |

Li's D = 1.39584 and Fu & Li's F = 1.40624) subgroups highly supported a sudden population contraction and/or balancing selection (Table 2).

Tajima' D results in overall negative values (Tajima' D = -1.83320; p<0.05) from both the tests that there is an excess of rare mutations in the subgroups, and the excess is statistically significant. The overall negative values (Fu's $F_S$ = -0.544) also resulted from Fu's $F_S$ test. However, the overall Fu & Li's D and Fu & Li's F (Fu & Li's D = 1.85839, p<0.02; Fu & Li's F = 0.58884) values were positive, and this revealed an excess of ancestral/prototype variants, that have been selected for in the past (Table 2). In other words, the number of unique variants present was low, and the ones that were present were carried by a large number of individuals.

Pairwise fixation index ($F_{ST}$) values among the different coinfected groups were assessed for measuring population differentiation based on their level of genetic differentiation. Pair-wise $F_{ST}$ values were also obtained from the comparison between specific co-infected/sole-infected subgroups, and these values were assessed to measure population differentiation. An $F_{ST}$ greater than 0.15 can be interpreted as very little gene flow and is significant in differentiating populations [43]. According to Table 3 shown, very little gene flow was obtained among the subgroups of Sole D with IEH, IOEP with IEH and the subgroups of IEH with ISTH. Consequently, very high genetic differentiation was observed in these subgroups. The highest $F_{ST}$ value was estimated between co-infected subgroup IEH against subgroups ISTH ($F_{ST}$ value = 0.34783) in all possible combinations of co-infected/Sole infected subgroups. The co-infected subgroups IOEP against ISTH exhibited the lowest $F_{ST}$ value ($F_{ST}$ value = 0.0000) (Table 3), indicating the highest level of gene flows between these two subpopulations with the shortest distance and the most increased accessibility. Overall the estimated genetic differentiation index was highly statistically significant ($X^2$ = 77.195, P<0.001) among the parasite subpopulations.

## Discussion

The study was conducted to determine the prevalence of *E. moshkovskii* in stool samples of diarrheal patients admitted to Infectious Disease Hospital, Kolkata and B C Roy Hospital, Kolkata. This is the first study done in Eastern India using molecular biological techniques that report the prevalence of *E. moshkovskii* in clinical stool samples. According to the active surveillance study for the detection of common enteric parasites going on over the past two decades, we have observed a decreasing trend of *E. histolytica* infection in the last few years around Kolkata [42,44–45]. Also, non-seasonal sporadic infections with *E. histolytica* have been observed, which is unusual for a tropical area like Kolkata [42,45]. Microscopic investigation of diarrheal stool samples has uncovered the presence of cysts/trophozoites of amoeba that have a similar morphological feature to *E. histolytica* by a significant proportion throughout the year in Kolkata. After performing PCR-based molecular identification, it was found that the most abundant species of *Entamoeba* observed in diarrheal stool samples in Kolkata is a morphologically indistinguishable amoeba from *E. histolytica* and is a related species called *E. moshkovskii*. A recent study in Egypt showed that 85% of amoebic infections were caused by

so-called non-pathogenic *Entamoeba* spp. such as *E. dispar*, *E. moshkovskii*, *and E. hartmani* [46]. This observation has led to an increased interest in the study of traditionally non-pathogenic *Entamoeba* species. Humans are a true host for amphizoic amoeba *E. moshkovskii* [27]. Moreover, recent evidence from different studies supports the pathogenicity of *E. moshkovskii* [23,27,29–30]. Although *E. moshkovskii* is identified as a cause of human infection, endemicity has not been appropriately assessed in most epidemiological studies [47]. In this study, we employed microscopic and molecular tools to determine the prevalence and genetic structure of *E. moshkovskii* in and around Kolkata.

The present study reported that *Entamoeba* spp was prevalent in diarrhoeal stool samples and other enteric pathogens in Kolkata. More than half of the amoebic infection was caused by *E. moshkovskii*. This finding is alarming as it implies that *E. histolytica* infections previously decreased and *E. moshkovskii* has been taking its place. Although *E. moshkovskii* was highly prevalent in diarrheal patients, we did not find any hematophagous trophozoites of *E. moshkovskii* during microscopy, indicating its non-invasive nature. These results will boost research for a better understanding of the mechanism of pathogenicity in the parasite. The raw data revealed that a more significant proportion of men were infected with *E. moshkovskii* than women. Still, the difference was not statistically significant, which indicates that the probability of infection with *E. moshkovskii* is equal for both genders. Our data revealed that *E. moshkovskii* infections were most predominant in ages 5–12. The higher prevalence of *E. moshkovskii* in this age group might be associated with their lack of health education regarding hygiene practices. A recent study in Bangladesh reported that 21% of children aged 2–5 were infected with *E. moshkovskii*, which was associated with diarrhoea [23]. Many other studies also reported *E. moshkovskii* as an entero-pathogen in patients suffering from diarrhoea or dysentery [7,23,27,29–30]. Our study was conducted among patients with diarrhoeal complaints. A notable percentage of individuals were infected with *E. moshkovskii*, and the presence of mono-infection/ sole infection of *E. moshkovskii* was statistically associated with diarrhoeal occurrence. Therefore, the diarrheal incidents associated with *E. moshkovskii* were not commonly coinfected in Kolkata. These results indicate that *E. moshkovskii* may not simply be a commensal of the human gut; instead, it acts as a "potential" pathogen causing diarrhoea and other gastrointestinal disorders in the study area.

A typical seasonal pattern generally observed in many parasitic infections like *E. histolytica* and *G. lamblia* usually showed the highest peaks in the wet season. It gradually decreased with the arrival of the dry season [48–52]. Interestingly we observed a unique seasonal pattern of *E. moshkovskii* infection in Kolkata. We reveal two remarkable peaks of infection in summer and post-fall season. The result of such an unusual study finding has yet to be explained, requiring further detailed investigations.

Further, we have performed phylogenetic analysis and multiple alignments of the *E. moshkovskii* population from Kolkata based on the 18S rRNA locus. Multiple alignments showed that 44.12% (n/N = 30/68) isolates were 100% similar at the sequenced region compared to the Laredo strain of *E. moshkovskii*, and the rest of the isolates were novel genetic variants. Phylogenetic analysis clarified the relationships among subpopulations clustered together with their respective variants. However, further research is needed using high-resolution molecular markers to conclude whether the subpopulation is a different genotype of *E. moshkovskii* or a completely new lineage. Since this was a hospital-based surveillance study and patients came from a limited geographical area, the distribution of different genotypes needed to be better understood. But the findings of this study confirmed the distribution of a richly diverse population of *E. moshkovskii* species in Kolkata and adjacent areas. The findings of this study highlighted the epidemiological significance of *E. moshkovskii* infection in Eastern India as it is the first report in this geographical area and exposes the existence of this species as a possible emerging enteric pathogen in India.

Most research on *E. moshkovskii* infection has mainly focused on the prevalence of this pathogen without considering the co-occurrence of other enteric pathogens [22–33]. In this study, a correlation was observed between 18S rRNA SNPs and clinical features. However, the correlation between SNPs and clinical features does not necessarily indicate a direct control of their impact on pathogenicity. The sequence of the 18S ribosomal RNA can be used as a phylogenetic marker, allowing for the identification of pathogenic organisms in clinical samples. It can also be utilized to detect *E. moshkovskii* isolates and enables further diagnostic testing. The identified SNPs may exhibit an essential role of *E. moshkovskii* in adapting to the gut environment or in acquiring other enteric pathogens.

The nucleotide diversity ($\pi$) value is an important index in molecular genetics to determine the degree of genetic polymorphism within a population [53]. The estimator of nucleotide diversity, $\pi$, with a higher value than 0.01, suggests comparatively significant variations in most organisms [54–55]. In this study, nucleotide diversity for all subgroups was lower than 0.01, indicating a lower degree of genetic polymorphism among the haplotypes. The haplotype diversity index was highest in the IB/V subgroup and lowest in the ISTH subgroup. The average haplotype diversity index was 0.677, considering differences among the haplotypes of each subgroup. The obtained average haplotype diversity index suggests that the *E. moshkovskii* population is highly diverse in this geographical area and influenced by a moderate recombination rate [53]. The high levels of genetic diversity suggested the strong viability and adaptability of the *E. moshkovskii* population. This may increase the average fitness of *E. moshkovskii* populations in a changing environment. However, the 18S rRNA data showed that the average nucleotide diversity was reasonably low. The nucleotide diversity was low since it is a highly conserved component with minimal nucleotide substitution rates. Our neutrality test results implied that the Sole D and IOEP subpopulations were not under directional selection pressure. IEH subpopulation was influenced by selection pressure, which resulted in the adaptation of these isolates to coexist with *E. histolytica* via changes in its genetic constitution. Pairwise genetic differentiation ($F_{ST}$) among different coinfected subgroups ranged from low to high. The $F_{ST}$ can range from 0 to 1, where 0 suggests complete sharing of genetic material and 1 suggests no sharing [53,55]. According to the standard $F_{ST}$ scale, the fixation index is $F_{ST}$ less than 0.05 = little genetic difference; $F_{ST}$ of 0.05 0.15 = moderate genetic difference; $F_{ST}$ of 0.15–0.25 = great genetic difference and $F_{ST}$ greater than 0.25 = very great genetic difference [43]. The majority of $F_{ST}$ values of different coinfected subgroups obtained in this study were lower than 0.25, indicating low to moderate genetic differentiation within the different coinfected/sole infected subgroups of the *E. moshkovskii* population in this geographical area. Whereas $F_{ST}$ values with the highest levels of differentiation were observed for IEH subgroups versus three other subgroups–Sole D, IOEP and ISTH indicating a higher genetic differentiation and higher genetic drift or lower gene flow within IEH coinfected subgroups of the parasite [55]. The higher $F_{ST}$ values also implied that this genome region may have undergone positive selection pressure in IEH coinfected subgroup [56]. Therefore, the obtained IEH subgroup might be genetically isolated and corresponds to the speciation process. More studies should be performed using other genetic markers to validate whether the coinfected subgroup of *E. moshkovskii* really corresponds to a new lineage or only to a different genotype of *E. moshkovskii*; it should be remembered that we only considered a fragment of 18S rRNA gene in this study.

## Conclusion

In conclusion, the present study suggested that *E. moshkovskii* is one of the causative agents for acute diarrhoea in humans. The study found that many diarrhoeal patients infected with this species were negative for other enteric pathogens such as bacteria and viruses. The study

recommends further research to understand the transmission dynamics of *E. moshkovskii* and proper diagnosis to avoid the development of drug-resistant strains. It also highlights the need for public health authorities to implement prevention and control strategies. The findings of the study raise concerns about the importance of proper diagnosis and control of *E. moshkovskii* infection.

## Supporting information

**S1 Table. List of clinical isolates of *E. moshkovskii* in diarrheal patients obtained from the hospital-based surveillance study.**
(DOCX)

**S2 Table. Substitution matrix based on 18S rRNA sequences of the populations of *E. moshkovskii* collected in and around Kolkata.**
(DOCX)

## Acknowledgments

The authors extend their appreciation to all patients who took part in the study. We are also grateful for the assistance provided by the hospital staff and sample collector in the sample collection process

## Author Contributions

**Conceptualization:** Sanjib K. Sardar, Sandipan Ganguly.

**Data curation:** Sanjib K. Sardar, Ajanta Ghosal, Tapas Haldar, Maimoon Maruf.

**Formal analysis:** Sanjib K. Sardar, Koushik Das, Sandipan Ganguly.

**Funding acquisition:** Sandipan Ganguly.

**Investigation:** Yumiko Saito-Nakano, Seiki Kobayashi, Sandipan Ganguly.

**Methodology:** Sanjib K. Sardar, Ajanta Ghosal, Tapas Haldar, Maimoon Maruf.

**Project administration:** Shanta Dutta.

**Supervision:** Sandipan Ganguly.

**Validation:** Koushik Das, Yumiko Saito-Nakano, Tomoyoshi Nozaki, Sandipan Ganguly.

**Visualization:** Seiki Kobayashi, Sandipan Ganguly.

**Writing – original draft:** Sanjib K. Sardar.

**Writing – review & editing:** Sandipan Ganguly.

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
