## [Decision Letter · Decision Letter 0]

23 Jan 2023

Dear Ganguly,

Thank you very much for submitting your manuscript "Prevalence and molecular characterization of Entamoeba moshkovskii in diarrheal patients from Eastern India" for consideration at PLOS Neglected Tropical Diseases. As with all papers reviewed by the journal, your manuscript was reviewed by members of the editorial board and by several independent reviewers. In light of the reviews (below this email), we would like to invite the resubmission of a significantly-revised version that takes into account the reviewers' comments. 

We cannot make any decision about publication until we have seen the revised manuscript and your response to the reviewers' comments. Your revised manuscript is also likely to be sent to reviewers for further evaluation.

Sincerely,

Maria Fantinatti, PhD

Guest Editor

Ricardo Fujiwara

Section Editor

Reviewer's Responses to Questions

**Key Review Criteria Required for Acceptance?**

**Methods**

-Are the objectives of the study clearly articulated with a clear testable hypothesis stated?

-Is the study design appropriate to address the stated objectives?

-Is the population clearly described and appropriate for the hypothesis being tested?

-Is the sample size sufficient to ensure adequate power to address the hypothesis being tested?

-Were correct statistical analysis used to support conclusions?

-Are there concerns about ethical or regulatory requirements being met?

Reviewer #1: Yes, the author did something, and described all contents in detail. And they did incomplete and be necessary to summarize really useful results from so many analyses.

Reviewer #2: The authors applies correct methodology, and the study design and materials are clearly laid out

**Results**

-Does the analysis presented match the analysis plan?

-Are the results clearly and completely presented?

-Are the figures (Tables, Images) of sufficient quality for clarity?

Reviewer #1: The manuscript contains a large number of figures and tables. Some of them contain little information， for example Fig 2, Fig 3 and Fig 4 could be deleted. Fig 5, Fig 6 and Fig 7 could be composed to one figure. Fig 10 to 13 could be composed to one figure. Data of table 1, table 2, Table 6 and Table 7 are duplicated with figures. Table 3 and table 5 could submit as a supplementary file. 

Some figures were of poor quality, such as Fig 5, Fig 6, Fig 10 to 13. The author should use professional software to reproduce.

Reviewer #2: the results are clearly and completely presented. However, figure 9 is with a low resolution

**Conclusions**

-Are the conclusions supported by the data presented?

-Are the limitations of analysis clearly described?

-Do the authors discuss how these data can be helpful to advance our understanding of the topic under study?

-Is public health relevance addressed?

Reviewer #1: Yes， the author did something， it was necessary to summarize really key conclusions from so many analyses.

Reviewer #2: The authors wrote a conclusion is too long and sounding repeatitive.

 The conclusion section is where auhtors a brief summary of the paper’s main points, but don’t simply repeat things that were in your paper. A conclusion is expressed in a few precisely worded sentences, usually one paragraph or 200 to 300 words and interpret your findings at a higher level of abstraction.

**Editorial and Data Presentation Modifications?**

Reviewer #1: I will suggest performing major Revision

Reviewer #2: The manuscript has important at local scale, and should be of great interest to the readers from Kolkata and adjacent areas, Eastern India. Findings are well presented and the manuscript present in an intelligible fashion but it is not written in standard English. 

I recommend that authors send their manuscript for professional copy editing and rewrite a conclusion 

The paper is recommended for Minor Revision

**Summary and General Comments**

Reviewer #1: The manuscript reported high prevalence rate of sole infection with E. moshkovskii in diarrhoeal patients in two local hospitals in India. The investigation involved a large number of clinical samples and demonstrated that E. moshkovskii is one of the causative agents for acute diarrhea in human，and indicated E. moshkovskii had pathogenic potential. It is an interesting and meaningful molecular epidemiological study，but the manuscript needed to be major revised.

1. The author described all contents in detail, but the main results and conclusions of the article were not highlighted. Manuscripts can be greatly simplified.

2. The manuscript contains a large number of figures and tables. Some of them contain little information， for example，Fig 2, Fig 3 and Fig 4 could be deleted. Fig 5, Fig 6 and Fig 7 could be composed to one figure. Fig 10 to 13 could be composed to one figure. Data of table 1, table 2, Table 6 and Table 7 are duplicated with figures. Table 3 and table 5 could submit as a supplementary file. 

3. Additionally, some figures were of poor quality, such as Fig 5, Fig 6, Fig 10 to 13. The author should use professional software to reproduce.

4. The author used many tools to analyze polymorphism of E. moshkovskii18S gene. The results seemed similar. In addition, figure 9 suggested high similarity in E. moshkovskii18S gene and the 18S gene was a conserved gene. It was necessary to summarize really useful results from so many analyses.

Reviewer #2: (No Response)

PLOS authors have the option to publish the peer review history of their article (what does this mean?). If published, this will include your full peer review and any attached files.

Reviewer #1: No

Reviewer #2: No
---

## [Decision Letter · Decision Letter 1]

10 Mar 2023

Dear Sandipan Ganguly,

Thank you very much for submitting your manuscript "Prevalence and molecular characterization of Entamoeba moshkovskii in diarrheal patients from Eastern India" for consideration at PLOS Neglected Tropical Diseases. As with all papers reviewed by the journal, your manuscript was reviewed by members of the editorial board and by several independent reviewers. The reviewers appreciated the attention to an important topic. Based on the reviews, we are likely to accept this manuscript for publication, providing that you modify the manuscript according to the review recommendations. 

Sincerely,

Maria Fantinatti, PhD

Guest Editor

Ricardo Fujiwara

Section Editor

Reviewer's Responses to Questions

**Key Review Criteria Required for Acceptance?**

**Methods**

-Are the objectives of the study clearly articulated with a clear testable hypothesis stated?

-Is the study design appropriate to address the stated objectives?

-Is the population clearly described and appropriate for the hypothesis being tested?

-Is the sample size sufficient to ensure adequate power to address the hypothesis being tested?

-Were correct statistical analysis used to support conclusions?

-Are there concerns about ethical or regulatory requirements being met?

Reviewer #1: Yes

Reviewer #2: There are no mention about methodology for virus and bacteria detections.

**Results**

-Does the analysis presented match the analysis plan?

-Are the results clearly and completely presented?

-Are the figures (Tables, Images) of sufficient quality for clarity?

Reviewer #1: Yes

Reviewer #2: (No Response)

**Conclusions**

-Are the conclusions supported by the data presented?

-Are the limitations of analysis clearly described?

-Do the authors discuss how these data can be helpful to advance our understanding of the topic under study?

-Is public health relevance addressed?

Reviewer #1: Yes

Reviewer #2: (No Response)

**Editorial and Data Presentation Modifications?**

Reviewer #1: The manuscript has been revised according to the comments.

Reviewer #2: (No Response)

**Summary and General Comments**

Reviewer #1: (No Response)

Reviewer #2: I summarize my comment below based on the general context

The authors must correct all types grammatical error in manuscript. For instance, in line 563, replace “dirrhoeal” by diarrhoeal

As mentioned by authors, in line 362 , the authors argued that “ Interestingly, 100% (n/N=17/17, 95% CI 0.784-363 1.00) of trophozoites were found to be non-hematophagous, indicating that E. moshkovskii may cause a non invasive form of intestinal amoebic infection”

This information is redundant because of globally it is observed that 90% of E. histolytica infections are asymptomatic and only in case of invasive amebiasis is observed hematophagous trophozoites on stool microscopy.

In line 319 , item Data collection of other Enteropathogen infections:

The authors mention: " Entamoeba histolytica, Cryptosporidium, and Giardia were identified by PCR method as described elsewhere. Helminth parasites were detected by light microscopy after wet mount. Other enteric pathogens like Vibrio cholera O1/O139, Salmonella spp., Campylobacter jejuni, Rotavirus, astrovirus and adenovirus co-infection information with entamoeba were obtained from the institutional database of ICMR-National Institute of Cholera and Enteric Diseases, Kolkata, India. Detailed information is available elsewhere [41]".

This is reference is about a hospital-based laboratory surveillance study that it was conducted among the patients admitted between November 2007 and October 2008 to the Infectious Diseases (ID) Hospital (Population = 1103) with diarrhoeal complaints.

Could the author explain about it. Where is detailed information about of present paper data (hospital-based systemic surveillance data conducted from March 2017 to February 2020)?

In line 374, diarrheagenic organisms were screened in the E. moshkovskii-positive diarrheal cases. There are no mention about methodology for virus and bacteria detections. What were methods employed?

In line 392 item GIS Mapping: The authors mentioned

“GIS mapping of the patients admitted to the I.D. hospital and B C Roy Children Hospital pointed out that the Beliaghata, Entally, Kashipur-Belgachia, Maniktala, Gopalpur-Rajarhat, Kolkata-Port, Jorasanko, Baruipurpurba, Maheshtala, Metiabruz, Bhangar areas of Kolkata and the adjacent regions were very highly infected (>3.5%) with Entamoeba moshkovskii. Ballygunge, Chowragee, Dumdum Uttar, Dumdum, Bidhannagar, Bhabanipur and Kashba had high prevalence rate (3%-3.5%). Behala Paschim, Behala Purba, Jadavpur were moderately (2.5%-2.99%) infected, while the lowest prevalence (< 2.49%) was reported in Baranagar, Rajarhat New Town and Shyampukur areas [Figure 3]. Those areas that constituted less than 1% of the total sample size have been excluded from the mapping”.

I suggest put prevalence rate in figure GIS Mapping and the authors must reduce this paragraph in order to avoid replicate information. Images help readers visualize the information.

In line 408, the authors mentioned “the rest of the 32 sequences represented genetic variants of E. moshkovskii not described earlier”

How many nucleotide sequences (genetic variants of E. moshkovskii ) are currently avaliable by GenBank?

intraspecific polymorphism exist but this information should be analysed based on robust data

In line 587 the authors mentioned that “SNPs in the 18S rRNA gene could have a significant impact on the biology of the amoeba”

Could the authors explain about it ? what is significant consequence ? 

 Could the authors place this sentence within the context of previous studies

The weaknesses of the study is centred in discussion section 

The authors should compare and contrast to previous studies. And also, although interpretation is the primary goal of the Discussion section, authors must be careful not to overinterpret their data, or stray too far from scientific evidence.

The author should providing proper context for your research and avoiding introducing new information

For instances, in line 592 “E. moshkovskii infection may alter susceptibility to infection with other enteric pathogens and modulate the effects of co-infecting gut pathogens”.

 In line 593 “E. moshkovskii may cause inflammation in the gut and create a niche that helps in the survival and proliferation of other enteropathogenic bacteria/viroses”. 

In line 594 “The observed multiple infections with many helminths and protozoan parasites could be explained by their same mode of transmission and poor hygiene areas. 

In line 596 “ Co-infection of E. moshkovskii and E. histolytica also suggests the absence of competitive exclusion among them. We also did not document any invasive disease in the diarrheal patients infected with E. moshkovskii”. 

In line 598 “More interestingly, the E. moshkovskii and E. histolytica co-infected subgroup also did not show any invasive disease indicating the parallel evolution of these two species. Therefore, the mystery is to be revealed under what conditions E. moshkovskii cells turn pathogenic, as many diarrhoeal incidences were observed that were solely infected with this species. This capability may result from particular parasite genetics and/or genotypes in E. moshkovskii isolates.

In general, the authors should providing proper context for your research and avoiding introducing new information and also add references

PLOS authors have the option to publish the peer review history of their article (what does this mean?). If published, this will include your full peer review and any attached files.

Reviewer #1: No

Reviewer #2: No

Figure Files:

Data Requirements:

Reproducibility:

References

---

## [Decision Letter · Decision Letter 2]

5 Apr 2023

Dear Ganguly,

We are pleased to inform you that your manuscript 'Prevalence and molecular characterization of Entamoeba moshkovskii in diarrheal patients from Eastern India' has been provisionally accepted for publication in PLOS Neglected Tropical Diseases.

Best regards,

Maria Fantinatti, PhD

Guest Editor

Ricardo Fujiwara

Section Editor

Reviewer's Responses to Questions

**Key Review Criteria Required for Acceptance?**

**Methods**

-Are the objectives of the study clearly articulated with a clear testable hypothesis stated?

-Is the study design appropriate to address the stated objectives?

-Is the population clearly described and appropriate for the hypothesis being tested?

-Is the sample size sufficient to ensure adequate power to address the hypothesis being tested?

-Were correct statistical analysis used to support conclusions?

-Are there concerns about ethical or regulatory requirements being met?

Reviewer #2: There are no new analyses/experiments required.

**Results**

-Does the analysis presented match the analysis plan?

-Are the results clearly and completely presented?

-Are the figures (Tables, Images) of sufficient quality for clarity?

Reviewer #2: The authors shoud increase resolution of figures and images

**Conclusions**

-Are the conclusions supported by the data presented?

-Are the limitations of analysis clearly described?

-Do the authors discuss how these data can be helpful to advance our understanding of the topic under study?

-Is public health relevance addressed?

Reviewer #2: (No Response)

**Editorial and Data Presentation Modifications?**

Reviewer #2: (No Response)

**Summary and General Comments**

Reviewer #2: Figures are an important part of research but The authors shoud increase resolution of figures and images in order to visually convey findings of article.

PLOS authors have the option to publish the peer review history of their article (what does this mean?). If published, this will include your full peer review and any attached files.

Reviewer #2: No

---

## [Editor Report · Acceptance letter]

8 May 2023

Dear Dr Ganguly,

We are delighted to inform you that your manuscript, "Prevalence and molecular characterization of Entamoeba moshkovskii in diarrheal patients from Eastern India," has been formally accepted for publication in PLOS Neglected Tropical Diseases.

Best regards,

Shaden Kamhawi

co-Editor-in-Chief

Paul Brindley

co-Editor-in-Chief
